# Median Nerve Stimulation for Treatment of Tics: A 4-Week Open Trial with Ecological Momentary Assessment

**DOI:** 10.3390/jcm12072545

**Published:** 2023-03-28

**Authors:** Ann M. Iverson, Amanda L. Arbuckle, David Y. Song, Emily C. Bihun, Kevin J. Black

**Affiliations:** 1Washington University School of Medicine, Washington University in St. Louis, St. Louis, MO 63110, USA; 2Department of Psychiatry, Washington University in St. Louis, St. Louis, MO 63110, USA; 3University of Rochester School of Medicine and Dentistry, Rochester, NY 14642, USA; 4Departments of Psychiatry, Neurology, Radiology and Neuroscience, Washington University in St. Louis, St. Louis, MO 63110, USA

**Keywords:** clinical trial, median nerve stimulation, open label, persistent motor tic disorder, tic disorders, Tourette syndrome

## Abstract

Median nerve stimulation (MNS) at 10–12 Hz was recently proposed as a treatment for Tourette syndrome and other chronic tic disorders (TS/CTD). We report on 31 participants ages 15–64 with TS/CTD in an open-label, comparative (within-group, several time points) study of MNS (ClinicalTrials.gov registration number NCT05016765). Participants were recruited from completers of a randomized controlled trial (RCT) of MNS and were given a transcutaneous electrical nerve stimulation (TENS) unit to use as desired for 12 Hz MNS for 4 weeks. Participants were instructed to complete surveys regarding tic symptoms and stimulation discomfort before and after stimulation, as well as twice daily when randomly prompted by text message. Participants also completed an extensive final survey. Twenty-seven participants completed the study. Median device use was 1.5 days per week and 50 min per day used. Tic frequency improved during MNS (mean improvement: 1.0 on a 0–5 scale, *p* < 0.001), as did tic intensity (mean improvement: 0.9, *p* < 0.001). Mean discomfort was mild (1.2 on a 3-point scale). In total, 21 participants (78%) planned to continue using the device. Participants’ results in this study did not correlate significantly with their results in the blinded RCT. We found MNS to improve tic frequency and intensity with minimal side effects.

## 1. Introduction

New treatments are needed for chronic tic disorders (CTD), including Tourette syndrome (TS), which affect approximately 1.4 million children and adults in the United States [1,2]. Patients with these disorders have one or more vocal or motor tics (or both in the case of Tourette syndrome) which have persisted for at least one year [3]. Comorbid conditions are also common in this population, specifically ADHD and OCD [4]. Whereas a variety of treatments can be used for tic disorders, including various pharmacological and behavioral therapies, many patients desire new treatment options [5]. For pharmacologic agents in particular, there is a relative lack of conclusive evidence as to the optimal treatment regimen [6]. The development of new treatments is made more difficult by the fact that the etiology and pathophysiology of TS/CTD is still not well understood. Neuroimaging studies of these patients have demonstrated impaired function of the cortical-striatal-thalamic-cortical circuit and abnormal neurotransmitter function, but further research is needed in this area [7].

Morera Maiquez and colleagues from Stephen Jackson’s lab at the University of Nottingham recently proposed a completely novel treatment idea involving indirect rhythmic electrical stimulation of the brain via peripheral stimulation of the median nerve [8]. The hypothesis was that increasing EEG power in a frequency range associated with motor inhibition would reduce tics. They demonstrated that 12 Hz stimulation of the median nerve (MNS) evoked synchronous contralateral EEG activity over the primary sensorimotor cortex and created small but statistically significant effects on the initiation of voluntary movements. Blinded video ratings showed significant reduction in tic number and tic intensity during 1-min stimulation blocks, and participants reported a significantly lower urge to tic. We recently replicated these results and extended them with longer stimulation blocks and a comparison condition intended to serve as an active placebo [9].

These results suggest that MNS may be a promising treatment for Tourette syndrome, but the real-world practicality of such a treatment has not been tested, as no studies have been published outside of the laboratory setting. In day-to-day use of such a device, patients would not only be exposed to much longer treatment duration but might also identify practical barriers to its use. We therefore conducted an open-label outpatient study of 12 Hz MNS with a portable TENS (transcutaneous electrical nerve stimulation) device to assess the tolerability, practicality and efficacy of MNS outside of a controlled setting. Whereas a randomized trial is technically possible, using the non-randomized format allows us to collect preliminary information on tolerability and practicality much more quickly. We included ecological momentary assessments (EMAs) of tic intensity and frequency, i.e., brief surveys of tic severity and device usage several times a day, to minimize recall bias and improve response validity [10].

## 2. Materials and Methods

We followed the CONSORT reporting suggestions adapted to non-randomized trials (see Appendix A) [11].

### 2.1. Ethics Approval

The Washington University Human Research Protection Office (IRB) approved this study (approval # 202109160). All participants provided written informed consent (those under age 18 provided written documentation of assent and a parent consented). The study used a device cleared by the FDA for pain relief, 501(k) premarket notification number K080661, and the IRB deemed this a non-significant-risk device study.

### 2.2. Companion Randomized Controlled Trial

The open-label study is an extension of a randomized controlled trial (RCT), of which the results are published separately [9]. Briefly, the RCT was a double-blind, randomized, crossover study that compared rhythmic MNS to arrhythmic MNS in people ages 15–64 with TS. Stimulation occurred in a laboratory setting in two sessions approximately one week apart. Participants were randomized to receive either rhythmic or arrhythmic stimulation on the first visit and the other on the second visit. Stimulation was delivered in both one-minute and five-minute blocks, interspersed with blocks without stimulation. In the RCT, participants were video recorded during stimulation sessions, and tic frequency and intensity were later rated by a trained rater blind to presence of stimulation, type of stimulation, block order, and visit number.

### 2.3. Study Design

The open-label study protocol was published prior to enrolling the first study participant, with trivial changes published by enrollment of the third participant [12]. Over a period of 4 weeks, participants selected whether and when to apply MNS, and they reported on tic intensity each time the device was turned on or off and at random times twice daily when prompted by a text message.

The planned sample size for the RCT from which participants for the open-label study were invited was 32, chosen for practical reasons of enrollment and cost, with the expectation that 32 participants would reasonably reflect the experience of adults and older teenagers who were willing to participate in a RCT for treatment of tics.

The study’s goals included determining the real-world usage and apparent utility of stimulation in people with chronic tics, determining momentary self-rated efficacy and side effects of stimulation, and testing whether a participant’s RCT results predicted their results in open use. We hypothesized that MNS stimulation would be effective in the treatment of tics and that results would correlate with those found in the RCT.

Participants were recruited between November 2021 and April 2022. All data collection was completed by June 2022.

### 2.4. Participants

Participants were invited from the 32 people who completed an in-person RCT, of which the results are reported separately [9]. Briefly, inclusion criteria for the controlled trial were age 15–64 inclusive at initial screening visit, current DSM-5 Tourette’s Disorder or Persistent (Chronic) Tic Disorder, and at least 1 tic per minute (on average) during the 5-min baseline video session on the first visit. Exclusion criteria included, among others, an implanted device that could be affected by electrical current, significant neurologic disease not counting TS (exceptions included febrile seizures or uncomplicated migraine), and any recent or planned change in somatic or psychotherapeutic treatment. All inclusion and exclusion criteria appear in the published RCT study protocol [13]. In total, 31 of the 32 RCT participants consented to participate in this open follow-on study.

Participants provided permission for their baseline information from the RCT to be utilized for this study well. These baseline measures included the Yale Global Tic Severity Scale (YGTSS) [14], Diagnostic Confidence Index [15], Adult Tic Questionnaire (ATQ) tic severity rating [16,17], Premonitory Urge for Tics Scale (PUTS) [18], the ADHD Rating Scale [19], and a self-report version [20] of the Yale-Brown Obsessive Compulsive Scale (Y-BOCS) Scale [21,22]. In addition, participants provided information about past medical and family history, as well as current treatments.

### 2.5. Intervention

Participants in the study were provided a transcutaneous electrical nerve stimulation (TENS) unit (TENS-7000, Roscoe Medical, Inc., Middleburg Heights, OH) and 1-inch-diameter circular adhesive carbon film electrodes (Syrtenty^®^ part number TSYR1000, Walpole, MA). Participants were instructed on proper use of the device and proper electrode attachment (anode centered over the median nerve with the distal edge at the distal wrist crease, cathode approximately 30 mm proximal center-to-center; see also [23]). Participants could select either wrist for MNS, but almost all chose the non-dominant hand. Simulation consisted of 200 µs square-wave pulses at 12 Hz. Participants were instructed to use the device as frequently as desired for as long as desired. At the first study visit, the stimulation threshold for each participant was identified as the minimum current setting at which a visible twitch of the abductor pollicis brevis was observed. Suprathreshold stimulation was encouraged, with an explanation that previous physiological and efficacy data used suprathreshold MNS, but participants selected the stimulation current each time they turned on the device.

### 2.6. Outcome Measures

The pre-specified primary outcome measures included time spent using the device (minutes per day and number of days per week used over the four-week period), how many participants self-reported they planned to continue using the stimulator after the study period ended, comparison of tic frequency and of tic intensity between “turning ON” and “turning OFF” surveys [14], mean discomfort while using stimulator (0–3 scale adapted from the CGI-I Efficacy Index [24]), and relationship to the results from the blinded study (correlation of change in tic frequency and intensity during this study with change in tic frequency and intensity respectively during active, rhythmic stimulation during the RCT). Tic intensity ratings used the 0–5 scale of the tic intensity item on the YGTSS [14]. Tic frequency was adapted from the YGTSS tic frequency item, since the original scale reports on frequency over a 1-week period but the present study inquires about tic frequency over periods of minutes to hours. The anchor points provided for frequency ratings varied depending on the duration of stimulation (e.g., one cannot usefully answer questions about tic frequency over the past hour if stimulation lasted only 10 min; see the form in ref. [12] for details. All other outcome measures were secondary.

Over the four-week study period, participants were asked to complete electronic ecological momentary assessment (EMA) surveys at each instance of three types of events: immediately prior to turning on the device, immediately after turning off the device, and in response to twice-daily texts at random times between 09:00 and 21:00. Surveys prior to turning on the device included current tic frequency and intensity, and surveys after turning off the device included the same questions in addition to questions regarding discomfort from stimulation, overall symptom improvements, and current device settings [14]. Text-prompted surveys were similar; if the stimulator was off when the text arrived, an additional question asked how long ago the stimulator was last used.

At the conclusion of the four-week study period, the participants completed a final survey that included the Adult Tic Questionnaire (ATQ) tic severity rating [16,17], Premonitory Urge for Tics Scale (PUTS) [18], Clinical Global Impression–Efficacy Index [24], their perception of duration of improvement after stimulation, and their plans to continue using the device. The survey also solicited open-ended comments about MNS and study participation. All surveys were completed online via RedCap at the location of the participant’s choosing.

### 2.7. Statistical Analysis

Participant data were collected and stored using REDCap [25]. Descriptive data are provided as mean ± standard deviation or median and IQR, depending on the normality of the data. Paired t-tests, or paired samples Wilcoxon tests as appropriate, tested most primary and secondary outcomes. Correlation analysis with Pearson’s or Spearman’s coefficients as appropriate was conducted for relevant comparisons. Sample size is provided for each analysis, as available sample size varied based on participant survey completion. Analysis was conducted using R Statistical Software [26,27].

## 3. Results

Individual participant data appear in ref. [12].

### 3.1. Participants

The baseline characteristics of all participants are shown in Table 1.

### 3.2. Outcomes

Results for pre-specified primary and secondary outcomes measures are included in Table 2 and Table 3 below, as well as in Figure 1, Figure 2, Figure 3, Figure 4 and Figure 5.

### 3.3. Usage Patterns

Median usage of the device was 1.5 days per week, with an average of 50 min of stimulation per day used (Table 2). However, usage varied widely among participants, with some participants reporting up to daily use many hours per day (Figure 1). Five participants in the study never filled out a survey indicating that they had turned the device on after the initial study visit at which they were trained on usage of the device and survey response, including one participant who did not fill out any EMA surveys.

### 3.4. Daily Survey Results

Based on the results of surveys filled out immediately prior to and following stimulation sessions, the mean improvements in tic frequency and intensity were 1.0 (*p* < 0.001) and 0.9 (*p* < 0.001), respectively (Table 2). Tic frequency improved, on average, from “3 = PRETTY OFTEN” to “2 = SOMETIMES” (anchor description varied based on the relevant time interval). Mean tic intensity improved from 3 (moderate) to 2 (mild).

Discomfort during stimulation was low, with a median score of 1 corresponding to “discomfort noticeable, but not severe enough to concern me or to turn it off.” There was no correlation between the stimulation amplitude or discomfort associated with stimulation and tic intensity or frequency (Table 2, Figure 2). However, the range of stimulation amplitudes used was limited, with the majority of participants choosing the same amplitude. An exploratory analysis compared discomfort and amplitude; there was a trend for those who used higher amplitude stimulation to experience more discomfort (Figure 2e).

### 3.5. Duration of Improvement in Symptoms after Stimulation Ends

Pre-specified analyses dictated comparing the results of participants’ responses to text surveys sent at random times. Surveys filled out within 60 min of the end of stimulation were compared with those filled out more than 60 min following the end of stimulation. There were no significant differences in tic frequency or intensity between the two groups, with mean differences of 0.1 (*p* = 0.20) and 0.3 (*p* = 0.77), respectively (Table 2). At the conclusion of the study, participants were surveyed on how long they felt benefits lasted after the conclusion of stimulation. The median response was 15 min (IQR 35), so a similar analysis to that listed above was conducted comparing responses received within 10 min (the nearest available time point in the survey) to those received after 10 min *(*Table 2). Again, there were no significant differences in tic frequency or intensity between the two groups, with mean differences of 0.4 (*p* = 0.30) and 0.3 (*p* = 0.47), respectively (Table 2). However, the sample size for this analysis was quite small (n = 7), as many participants did not have a random text response within 15 min of the end of a stimulation session.

### 3.6. Final Survey Results

In the final survey (completed by 27 of 31 participants), a majority of participants (78%) indicated that they planned to continue using the stimulator. Two-thirds (66%) indicated that their symptoms overall were either much improved or minimally improved on the CGI-I rating scale, with only a single participant who indicated their symptoms worsened during the study period (Figure 3).

The participants’ responses on the CGI-I scale were correlated with their frequency and intensity EMA scores during the study (r = −0.54 and r = −0.41) (Table 2, Figure 4). There were no significant changes in participants’ PUTS or ATQ scores from the beginning of the study period to the end, with mean differences of 0.7 (*p* = 0.84) and −0.9 (*p* = 0.85), respectively (Table 2).

Participants’ responses in this open-label study did not correlate significantly with their results on the rhythmic stimulation day of the RCT: r = 0.21 (*p* = 0.33) and r = 0.36 (*p* = 0.08) (Table 2, Figure 5).

### 3.7. Responders Analysis

Participants who had a CGI-I score of 3 (minimally improved) or better were categorized as responders to stimulation. Examining multiple factors, we did not find any statistically significant differences between participants who responded to the stimulation and those who did not (Table 3).

### 3.8. Open-ended Comments and Adverse Events

Participants were also asked a series of open-ended questions about the device. Full responses can be viewed in ref. [12].

The most common complaints were related to the form of the device. Many people felt that the TENS unit, which had long wires and adhesive pads that many participants removed after each use of the device, was too bulky or cumbersome to deal with on a regular basis. People frequently noted that they only used the device at home due to embarrassment or impracticality and suggested that a smaller design would improve their experience. There were also complaints that the amplitude dial was too sensitive, such that small position changes would produce big changes in the intensity of stimulation experienced. Some noted that correct positioning of the pads and of the wrist was crucial for good effect.

No serious adverse events occurred. In general, comments about discomfort ranged dramatically, from no discomfort to discomfort that heavily outweighed any perceived benefit. Two participants reported skin irritation at the electrode site, possibly contact dermatitis due to the adhesive [28], and two people felt that muscle activation with active stimulation limited the dexterity of their hands. Overall, many people suggested that they planned to use the device, but only for settings in which their tics were particularly bad. On the other hand, one person classified the device as “a godsend”.

### 3.9. Diurnal Variation of Tics

As an exploratory analysis, given the lack of prior EMA studies in TS, we examined diurnal effects on tic severity. Tic frequency and intensity without stimulation were relatively constant at different points throughout the day, with little variation between the hours of 9 a.m. and 9 p.m. when most participants supplied data (Figure 6).

## 4. Discussion

In this pilot study, median nerve stimulation significantly improved tic frequency and intensity with relatively low amounts of discomfort. Most participants planned to continue using the device following the conclusion of the study period. Our results are consistent with the hypothesis that the device has promise as a treatment for TS/CTD in a real-world setting.

Most participants did not use the device every day, with median usage of 1.5 days per week and 50 min per day of use. Participants’ open-ended comments helped to explain this frequency, with many participants stating that the format of the device with long wires and adhesive pads made it awkward to wear outside the home. Therefore, many participants stated that they planned to use the device only during periods where their tics were particularly bothersome or when they especially did not want their tics to be noticed.

Our study demographics were relatively representative of the general TS population, although the mean age was 34.5, while most people with TS who seek clinical care are children and adolescents [2]. The broad age range for our study was chosen to allow for study of the device in teenagers and adults of a variety of ages, as participants of different ages may respond to stimulation differently. Younger children were excluded, as some children under age 15 had withdrawn from a previous study of median nerve stimulation due to the sensation associated with stimulation [8]. Therefore, further study is needed in children. However, tic severity in our study was representative of TS patients seeking clinical care, with a median YTGSS total tic score of 25 compared with a mean of 24 in a recent large sample that included clinical trial participants and outpatients at referral centers [29].

As noted above, results for intensity, frequency, and discomfort from surveys throughout the study period were encouraging, with mean improvements of approximately one point in intensity and frequency, and mean discomfort of one on a 0–3 scale. In addition, these results are fairly consistent with participant reports on similar questions asked during the final surveys, and there was statistically significant correlation between the EMA surveys during the study and the CGI-I score on the final survey. This internal consistency is encouraging with regard to the validity of our results. ATQ and PUTS scores did not change from the beginning to the end of the study, which may relate to the fact that most patients used the device only intermittently and reported benefit persisting for, at most, a few minutes after stimulation was turned off.

The lack of significant correlation between the RCT and open study results was surprising. It is possible that this result arises from differences in participant self-reporting versus the blinded, objective clinical rating that was completed in the RCT. Alternatively, details of ad lib usage chosen by individual participants (e.g., pad placement, left vs. right forearm, amplitude) likely differed at times from the methods used in the RCT. We did not identify any characteristics that made a participant more likely to respond to stimulation, including the discomfort they experienced during stimulation or the amplitude they chose. A larger sample may be needed to detect such a difference; alternatively, different predictive factors may need to be examined.

The first MNS report in TS suggested that the benefits of stimulation might extend past the active stimulation period [8]. We did not detect post-stimulation improvement in our study, comparing tic frequency and intensity at 15 and 60 min after the end of stimulation. However, since random texts only occasionally reached participants shortly after they turned off the device, power to detect such improvement was limited, and this conclusion must be viewed as tentative.

To the best of our knowledge, this report also includes the first EMA data in a clinical trial in TS. We did not observe variation in tic frequency throughout daytime hours, though the power to detect such effects is limited. More importantly, EMA surveys during the trial period correlated with the CGI-I score at its end, supporting the validity of the EMA survey severity items chosen. However, EMA participation was variable, with a few participants completing no prompted surveys, and many responses were delayed more than a few minutes after the intended time. A more integrated smart phone app may improve response speed or completion rate.

### 4.1. Limitations

As mentioned above, our study is limited by a relatively small sample size. In addition, there were some participants for whom device usage and survey completion were limited, and it is possible that these participants would have had a more negative view of the device. Thus, while we were able to draw interesting conclusions about the utility of such a device, we were less able to definitively determine results for some of our secondary analyses such as the length of stimulation improvement and which participant characteristics predicted response to stimulation.

In addition, our study relied on participant self-reporting of tic frequency and intensity rather than more objective measures such as videotaping and clinician ratings. Although participant results were generally consistent between the EMA responses during the study and the end-of-study surveys, participants’ perception of their own tics may differ from what objective measurements of tics would show.

Finally, although participants were trained on the use of the device at the beginning of the study period, we have no independent way of confirming that the participants used the device as recommended after that point.

Repeating this study with a larger sample size would provide more support regarding our primary findings and more power to detect differences in our secondary analyses. In addition, a newer, less cumbersome device could be tested with a similar study design. If there is interest in further characterizing how long improvement lasts after stimulation, having participants fill out surveys at multiple timepoints after stimulation would provide significantly more data.

### 4.2. Conclusions

Overall, the results of this study are in line with the previously published RCT results from the University of Nottingham [8]. In that open, laboratory-based study, 10 Hz rhythmic MNS stimulation produced notable improvements in tic frequency and intensity in 6 one-minute stimulation blocks. The present results support the real-world utility and acceptability of MNS to patients. Generally, these results regarding efficacy are in line with the results from the RCT in the same participants, but this open-label study identified additional barriers to daily use of the device in practice.

### 4.3. Clinical Significance

Wearable wristwatch-style stimulation devices are being developed, which may address some complaints from patients that the TENS units were cumbersome. Ultimately, a smaller, wearable device would appear to be a valuable treatment option for many patients, including those who have failed more conventional treatments or those who do not prefer pharmacologic or behavioral therapies.

## Figures and Tables

**Figure 1 jcm-12-02545-f001:**
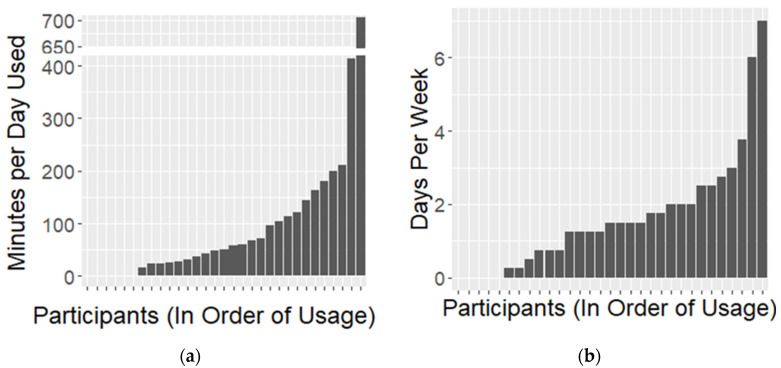
Participant Usage Patterns. (**a**) Minutes per day used by each participant. Participants are ordered by usage (n = 31). (**b**) Days per week for each participant. Participants are ordered by usage (n = 31).

**Figure 2 jcm-12-02545-f002:**
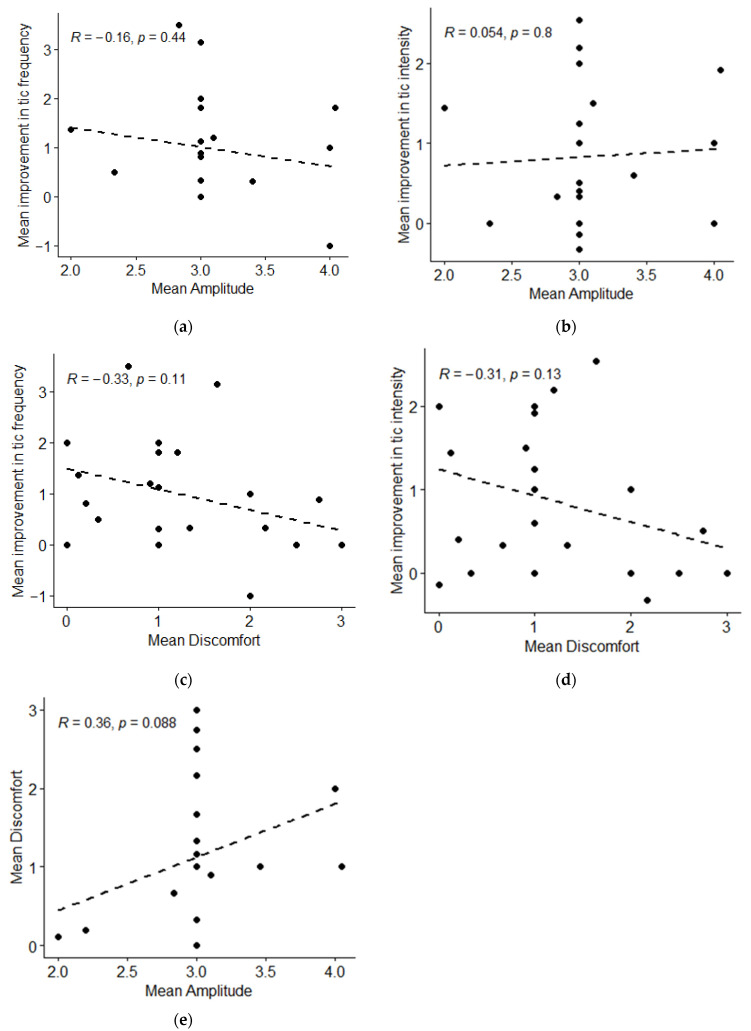
Impact of Stimulation Amplitude and Discomfort on Tic Frequency and Intensity. (**a**) Mean improvement in tic frequency compared with mean stimulation amplitude for each participant (n = 24). (**b**) Mean improvement in tic intensity compared with mean stimulation amplitude for each participant (n = 24). (**c**) Mean improvement in tic frequency compared with mean stimulation discomfort for each participant (n = 25). (**d**) Mean improvement in tic intensity compared with mean stimulation discomfort for each participant (n = 25). (**e**) Mean discomfort compared with mean amplitude for each participant (exploratory analysis) (n = 24).

**Figure 3 jcm-12-02545-f003:**
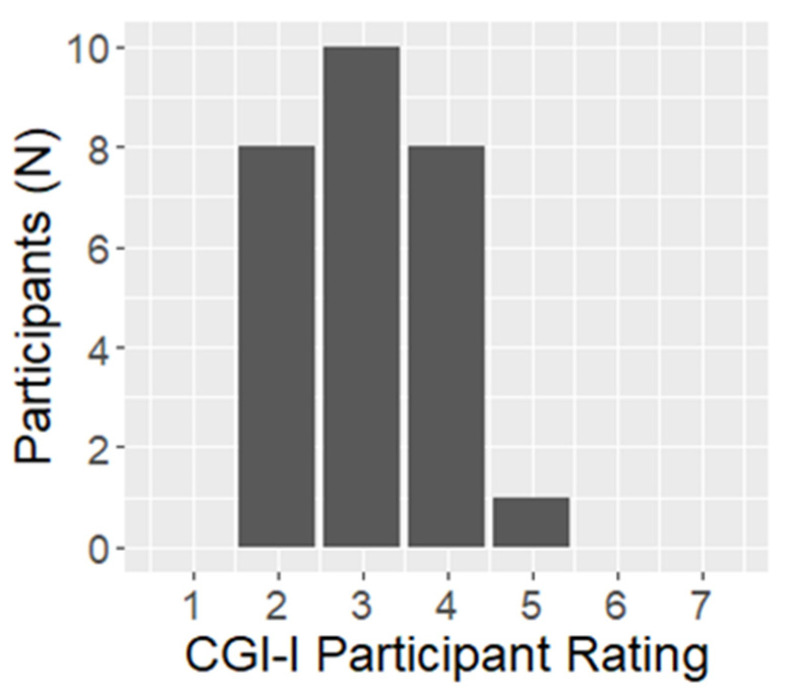
Final Participant Clinical Global Impression–Improvement Scores (n = 27). 1 = Very much improved—nearly all better; good level of functioning; minimal symptoms; represents a very substantial change. 2 = Much improved—notably better with significant reduction of symptoms; increase in the level of functioning but some symptoms remain. 3 = Minimally improved—slightly better with little or no clinically meaningful reduction of symptoms; represents very little change in basic clinical status, level of care, or functional capacity. 4 = No change—symptoms remain essentially unchanged. 5 = Minimally worse—slightly worse but may not be clinically meaningful; may represent very little change in basic clinical status, level of care, or functional capacity. 6 = Much worse—clinically significant increase in symptoms and diminished functioning. 7 = Very much worse—severe exacerbation of symptoms and loss of functioning.

**Figure 4 jcm-12-02545-f004:**
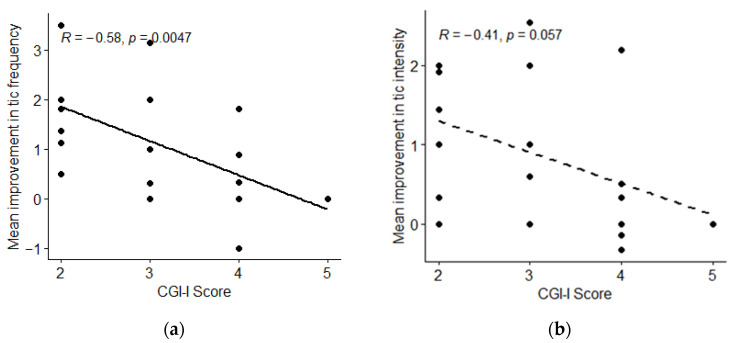
Correlation Between Participants’ EMA Responses During the Study and Their Report of Improvement During the Final Survey. (**a**) Mean improvement in tic frequency from EMA surveys compared with Clinical Global Impression–Improvement score for each participant (n = 22). (**b**) Mean improvement in tic intensity from EMA surveys compared with Clinical Global Impression–Improvement for each participant (n = 22).

**Figure 5 jcm-12-02545-f005:**
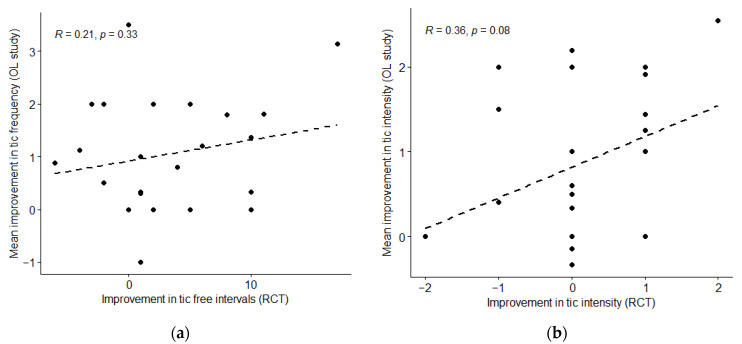
(**a**) Each participant’s mean improvement in tic frequency following stimulation sessions during this open-label study was compared with his or her improvement in tic frequency (number of 10-s tic-free intervals) in the first 5-min block of rhythmic stimulation in the RCT (n = 24). (**b**) Each participant’s mean improvement in tic intensity following stimulation sessions during open-label study was compared with his or her improvement in tic intensity in the first 5-min block of rhythmic stimulation in the RCT (n = 25).

**Figure 6 jcm-12-02545-f006:**
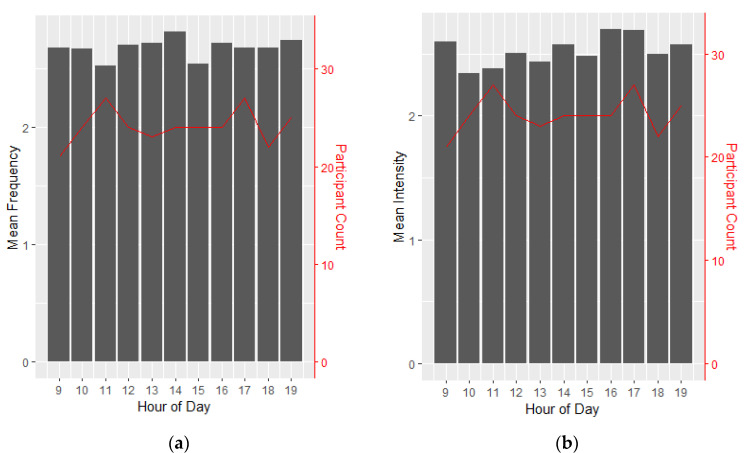
Tic Frequency and Intensity Throughout the Day. (**a**) Mean tic frequency per hour, with number of participants with results in each hour shown on right axis. Frequency is drawn from surveys filled out before turning on device and random surveys where device was not on. (**b**) Mean tic intensity per hour, with number of participants with results in each hour shown on right axis. Intensity is drawn from surveys filled out before turning on device and random surveys where device was not on.

**Table 1 jcm-12-02545-t001:** Baseline characteristics of all participants (n = 31).

	Mean	SD	Median	IQR
Age	34.5	16.6	35	16–45
Sex (male)	64.5%	NA	NA	NA
YGTSS Impairment	19.2	16.1	20	0–30
YGTSS Total Tic Score	24.9	9.1	25	20–29.5
DCI	60.9	20.2	56	44.5–78.5
ATQ	38.5	21.4	35	21.5–55
PUTS	21.7	7.2	23	15.5–27
Distress/Impairment Last Week	61.3%	NA	NA	NA
Distress/Impairment Lifetime	100%	NA	NA	NA
Current Antipsychotic Use	22.6%	NA	NA	NA
Current Alpha-2 Agonist Use	9.7%	NA	NA	NA
Lifetime Antipsychotic Use	35.5%	NA	NA	NA
Lifetime Alpha-2 Agonist Use	51.6%	NA	NA	NA
Number of Treatments Tried	3.9	3.8	3	1.0–5.5
Successful Treatments	0.6	1.0	0	0–1
Adequate Behavior Therapy	9.7%	NA	NA	NA
% of Participants with Family History of Tics in First Degree Relatives	45.2%	NA	NA	NA
% of Participants with Family History of Tics, OCD, or ADHD in First Degree Relatives	67.7%	NA	NA	NA
Y-BOCS Obsession Subtotal	3.5	3.7	3	0–5.5
Y-BOCS Compulsion Subtotal	3.0	2.9	2	0–6
Y-BOCS Total	6.5	5.9	5	0–11
ADHD Rating Scale	13.2	13.3	7	2.5–20.5

Baseline characteristics were gathered at the first study visit of the randomized controlled trial from which the open-label study participants were drawn. Y-BOCS = Yale-Brown Obsessive Compulsive Scale, PUTS = Premonitory Urge for Tics Scale, YGTSS = Yale Global Tic Severity Score, DCI = Diagnostic Confidence Index, ATQ = Adult Tic Questionnaire, OCD = Obsessive-Compulsive Disorder, ADHD = Attention-Deficit Hyperactivity Disorder. Successful treatments = number of treatments attempted judged by the participant to provide at least minimal improvement. Family history items reflect the percentage of first-degree relatives with the condition named.

**Table 2 jcm-12-02545-t002:** Outcomes.

Outcome	Result	95% C.I.	*p*	Type
Days per week device used (n = 31)	1.5, 1.4 (median, IQR)	1.1–2.3	—	primary
Minutes per day used (n = 31)	49.6, 93.0 (median, IQR)	45.5–150.3	—	primary
Number who plan to continue using device after 4 weeks (n = 27)	21 yes (77.8%), 6 no	62.1–93.4	—	primary
Change in tic frequency when turning device OFF versus previous ON (n = 25 for t-test)	Turning off: 2.1 ± 1.1 (mean ± SD) Turning on: 3.1 ± 1.0 (mean ± SD) Difference: 1.0 (mean)	0.6–1.4	<0.001 (paired *t* test)	primary
Change in tic intensity when turning device OFF versus previous ON (n = 25 for *t*-test)	Turning off: 2.0 ± 1.1 (mean ± SD) Turning on: 2.9 ± 0.8 (mean ± SD) Difference: 0.9 (mean)	0.6–1.9	<0.001 ^a^	primary
Mean discomfort while using stimulator (n = 26)	1, 0.9 (median, IQR)	1.0–1.7	—	primary
Effect on tic frequency compared to in RCT (see Figure 5a) (n = 24)	*r* = 0.21	−0.22–0.56	0.33	primary
Effect on tic intensity compared to in RCT (see Figure 5b) (n = 25)	*r* = 0.36	−0.05–0.66	0.08	primary
Overall impact of stimulation on symptoms throughout study period (n = 27)	3.1 ± 0.9 (mean ± SD)	2.7–3.4	—	secondary
Discomfort from final survey (n = 27)	1.2 ± 1.0 (mean ± SD)	0.8–1.6	—	secondary
Therapeutic effect while using from final survey (n = 27)	1.5 ± 0.9 (mean ± SD)	1.1–1.9	—	secondary
Difference in ATQ before and after study (n = 25)	0.7 ± 18.0 (mean ± SD)	−6.7–8.2	0.84	secondary
Difference in PUTS before and after study (n = 25)	−0.9 ± 4.3 (mean ± SD)	−2.8–1.0	0.35	secondary
Correlation between tic frequency and stimulation amplitude (see Figure 2a) (n = 24)	R = −0.16 Rho = −0.19	−0.5–0.26	0.44 0.37	secondary
Correlation between tic intensity and stimulation amplitude (see Figure 2b) (n = 24)	R = 0.05 Rho = 0.15	−0.36–0.45	0.80 0.49	secondary
Correlation between tic frequency and stimulation discomfort (see Figure 2c) (n = 25)	R = −0.33 Rho = −0.36	−0.64–0.08	0.11 0.08	secondary
Correlation between tic intensity and stimulation discomfort (see Figure 2d) (n = 25)	R = −0.31 Rho = −0.27	−0.63–0.09	0.13 0.19	secondary
Correlation between stimulation amplitude and stimulation discomfort (see Figure 2e) (n = 24)	R = 0.36 Rho = 0.31	−0.06–0.66	0.09 0.13	exploratory
Change in tic frequency when device OFF for more than 60 min vs. less than 60 min (*t* test, n = 16)	More than 60: 2.7 ± 0.9 (mean ± SD) Less than 60: 2.3, 2.1 (median, IQR) Difference: 0.3 (mean)	−0.2–0.7	0.20	secondary
Change in tic intensity when device OFF for more than 60 min vs. less than 60 min (*t* test, n = 16)	More than 60: 2.5 ± 0.8 (mean ± SD) Less than 60: 2.3, 2.2 (median, IQR) Difference: 0.1 (mean)	−0.4–0.5	0.77	secondary
Participants’ perception of length of tic improvement (n = 21)	15, 35 (median, IQR)	15–60	—	secondary
Change in tic frequency when device OFF for more than 10 min vs. less than 10 min (n = 7 for *t* test)	More than 10: 3, 2 (median, IQR) Less than 10: 2.2, 1.3 (median, IQR) Difference: 0.4 (mean)	−0.5–1.3	0.30	secondary
Change in tic intensity when device OFF for more than 10 min vs. less than 10 min (n = 7 for *t* test)	More than 10: 3, 2.5 (median, IQR) Less than 10: 1.9, 1.3 (median, IQR) Difference: 0.3 (mean)	−0.7–1.3	0.47	secondary
Correlation between tic frequency and CGI-I (see Figure 4a) (n = 22)	R= −0.58 Rho= −0.64	−0.80–−0.21	0.005 0.001	secondary
Correlation between tic intensity and CGI-I (see Figure 4b) (n = 22)	R= −0.41 Rho = −0.44	−0.71–0.01	0.06 0.04	secondary
Difference in days per week device used between RCT responders ^b^ (n = 13) and non-responders (n = 18)	RCT responders: 133.4 ± 207.9 (mean, SD) RCT non-responders: 72.3 ± 62.0 (mean, SD)	−66.9–189.0	0.32	exploratory
Difference in minutes per day used between RCT responders ^b^ (n = 13) and non-responders (n = 18)	RCT responders: 2.0 ± 2.3 (mean, SD) RCT non-responders: 1.4 ± 0.9 (mean, SD)	−0.9–2.0	0.44	exploratory

^a^ Paired samples Wilcoxon test. ^b^ RCT responders defined as those with CGI-I Score of 2 (much improved) or better on the rhythmic stimulation day.

**Table 3 jcm-12-02545-t003:** Responders Analysis.

	Mean, Non-Responders (n = 9)	Mean, Responders (n = 18)	*p* Value
Age	37.7	33.9	0.52
Y-BOCS Obsession Subtotal	2.7	4.2	0.45
Y-BOCS Compulsion Subtotal	2.9	3.3	0.90
Y-BOCS Total	5.6	7.6	0.51
PUTS	19.4	21.9	0.45
YGTSS Impairment	20.6	15.6	0.52
YGTSS Total Tic Score	22.9	25.8	0.12
DCI	52.7	64.3	0.20
ATQ	34.8	41.2	0.43
ADHD	9.1	14.4	0.20
Number of Treatments Tried	3.9	4	0.73
Successful Treatments	0.3	0.8	0.44
Family History of Tics (% of participants with history in first degree relative)	55.6	27.8	0.22
Family History Tics/OCD/ADHD (% of participants with history in first degree relative)	77.8	55.6	0.41
Sex (% male)	66.7	72.2	1
Distress/Impairment Last Week (% yes)	66.6	66.6	1
Distress/Impairment Lifetime (% yes)	100	100	1
Adequate Behavior Therapy (% yes)	0	16.7	0.53

Comparison of responders vs. non-responders, where responders are those with a CGI-I score of 3 (minimally improved) or better. Y-BOCS = Yale-Brown Obsessive Compulsive Scale, PUTS = Premonitory Urge for Tics Scale, YGTSS = Yale Global Tic Severity Score, DCI = Diagnostic Confidence Index, ATQ = Adult Tic Questionnaire. Successful treatments = number of treatments attempted rated by participant to provide at least minimal improvement. Family history items displayed as percentage of first-degree relatives with condition.

## Data Availability

Individual subject data is available via the Open Science Foundation page for the project in ref. [12].

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
