# Peer review of "Median Nerve Stimulation for Treatment of Tics: A 4-Week Open Trial with Ecological Momentary Assessment"

_jcm, 2023, doi:10.3390/jcm12072545_

Round 1

Reviewer 1 Report

The present research article by Iverson, entitled ‘Median nerve stimulation for treatment of tics. 2. A 4-week, open trial with ecological momentary assessment’ is a well-written and useful summary on the current status of knowledge on the possible application of median nerve stimulation as a promising treatment for Tourette syndrome. 

The main strength of this manuscript is that it addresses an interesting and timely question, conducting a study of stimulation of median nerve with a portable TENS (transcutaneous electrical nerve stimulation) device to assess the tolerability, practicality and efficacy in patients with Tourette syndrome.

In general, I think the idea of this article is really interesting and the authors’ fascinating observations on this timely topic may be of interest to the readers of Journal of Clinical Medicine. However, some comments, as well as some crucial evidence that should be included to support the author’s argumentation, needed to be addressed to improve the quality of the manuscript, its adequacy, and its readability, in particular reshaping parts of the Introduction and Methods sections by adding more evidence and theoretical constructs.

Please consider the following comments:

Abstract: According to the Journal’s guidelines, the abstract should be a total of about 200 words maximum. Also, this section should be presented as a single paragraph, without explicit sub-headings. Please correct the actual one.

A graphical abstract that will visually summarize the main findings of the manuscript is highly recommended.

In general, I recommend authors to use more references to back their claims, especially in the Introduction of this article, which I believe is lacking. Thus, I recommend the authors to attempt to expand the topic of their article, as the bibliography is too concise. Nevertheless, I believe that less than 50 articles are too low for a research article. Therefore, I suggest the authors to focus their efforts on researching relevant literature: in my opinion, adding more citations will help to provide better and more accurate background to this study. 

Introduction: The ‘Introduction’ section is well-written and nicely presented, with a good balance of descriptive text and information about etiology and symptomathology that define neurodegenerative disorders, in particular Tourette syndrome. Nevertheless, I believe that more information about pathophysiology and core features of these disorders will provide a better and more accurate background, because as it stands, this information is not highlighted in the text. In this regard, I would suggest to add more information on pathological neural substrates of neurodegeneration in Tourette syndrome, for example focusing on the role of specific brain areas, like prefrontal cortex, in the pathophysiology of this disorder. In this regard, I would suggest to add more information on pathological neural substrates of these disorders, for example focusing on ‘Dissecting Neurological and Neuropsychiatric Diseases: Neurodegeneration and Neuroprotection’ and on structural as well as functional abnormalities of prefrontal cortex that may affect patients’ cognitive impairments (https://doi.org/10.3390/biomedicines10123189). In my opinion, authors could further explore relationship between the molecular regulation of higher-order neural circuits and neuropathological alterations in this neuropsychiatric disorder (https://doi.org/10.3390/biomedicines10081897), in order to provide more insights on pathophysiological features of this disorder.

Participants: Could the authors specify how did they estimate the exact number of participants and provide more information about the diagnostic tests used for clinical evaluation?

Results: I suggest rewriting this section more accurately. To properly present experimental findings, I think that authors should provide full statistical details (like degree of freedom or post-hoc utilized), to ensure in-depth understanding and replicability of the findings. Also, in my opinion, it is necessary for the authors to present their findings using summary tables.

In my opinion, I think the ‘Conclusions’ and ‘Limitations’ paragraphs would benefit from some thoughtful as well as in-depth considerations by the authors, because as it stands, they are very descriptive but not enough theoretical as discussions should be. Authors should make an effort, trying to explain the theoretical implication as well as the translational application of their research.

References: Authors should consider revising the bibliography, as there are several incorrect citations. Indeed, according to the Journal’s guidelines, they should provide the abbreviated journal name in italics, the year of publication in bold, the volume number in italics for all the references. 

I hope that, after these careful revisions, this paper can meet the Journal’s high standards. 

I am available for a new round of revision of this paper. I declare no conflict of interest regarding this manuscript. 

Best regards,

Reviewer

Author Response

Response to Reviewer 1 Comments

We thank the reviewer for carefully considering this contribution.

Point 1: Abstract: According to the Journal’s guidelines, the abstract should be a total of about 200 words maximum. Also, this section should be presented as a single paragraph, without explicit sub-headings. Please correct the actual one.

Response 1: We have adjusted our abstract to be less that 200 words. We do not include explicit subheadings.

Point 2: A graphical abstract that will visually summarize the main findings of the manuscript is highly recommended.

Response 2: We have included a graphical abstract.

Point 3: In general, I recommend authors to use more references to back their claims, especially in the Introduction of this article, which I believe is lacking. Thus, I recommend the authors to attempt to expand the topic of their article, as the bibliography is too concise. Nevertheless, I believe that less than 50 articles are too low for a research article. Therefore, I suggest the authors to focus their efforts on researching relevant literature: in my opinion, adding more citations will help to provide better and more accurate background to this study.

The ‘Introduction’ section is well-written and nicely presented, with a good balance of descriptive text and information about etiology and symptomathology that define neurodegenerative disorders, in particular Tourette syndrome. Nevertheless, I believe that more information about pathophysiology and core features of these disorders will provide a better and more accurate background, because as it stands, this information is not highlighted in the text. In this regard, I would suggest to add more information on pathological neural substrates of neurodegeneration in Tourette syndrome, for example focusing on the role of specific brain areas, like prefrontal cortex, in the pathophysiology of this disorder. In this regard, I would suggest to add more information on pathological neural substrates of these disorders, for example focusing on ‘Dissecting Neurological and Neuropsychiatric Diseases: Neurodegeneration and Neuroprotection’ and on structural as well as functional abnormalities of prefrontal cortex that may affect patients’ cognitive impairments (https://doi.org/10.3390/biomedicines10123189). In my opinion, authors could further explore relationship between the molecular regulation of higher-order neural circuits and neuropathological alterations in this neuropsychiatric disorder (https://doi.org/10.3390/biomedicines10081897), in order to provide more insights on pathophysiological features of this disorder.

Response 3: We appreciate the feedback that our introduction could be expanded, and we have included additional information about the pathophysiology and core features of chronic tic disorders, including additional citations. We emphasize that this paper is a companion to the RCT report, which cites more background literature. We do find the neurobiological and neuropyschological underpinnings of different disorders to be important areas of study and worthy of future exploration. However, we feel that discussion of these concepts in any real depth is outside the scope of this (non-review) paper, which focuses primarily on a novel potential treatment for chronic tic disorders. We cite sources for all key assertions, methods, and any prior similar work (of which there is none). Beyond that point, whether any additional citations are appropriate appears to be a matter of taste. Not to argue, but simply to support that assertion, I’ll point out that in 30 years, I have never heard of 50 or any other ideal number of citations, except that many well-known journals specifically limit the number of citations allowed. The journal or editor may have their own opinion.

Point 4:: Participants: Could the authors specify how did they estimate the exact number of participants and provide more information about the diagnostic tests used for clinical evaluation?

Response 4: We planned in advance (see pre-registered protocol) to enroll every consenting participant who had completed the companion randomized, controlled trial. The RCT enrolled 32, which was the target for that trial. The present open-label study enrolled 31 participants from the 32 completers of the preceding RCT. Of those, 27 participants completed the final survey. For each analysis, we included all participants for whom the information necessary for that analysis was present. For example, for the analysis of “Change in tic frequency when turning device OFF versus previous ON (n=25 for t-test),” we included all participants who had at least one ON-OFF survey pair. For each analysis, we included the number of participants for that analysis.

              We have ensured that all diagnostic scales are cited appropriately in the methods section. They are located in both the participants section and in the outcomes measure section.

Point 5: Results: I suggest rewriting this section more accurately. To properly present experimental findings, I think that authors should provide full statistical details (like degree of freedom or post-hoc utilized), to ensure in-depth understanding and replicability of the findings. Also, in my opinion, it is necessary for the authors to present their findings using summary tables

Response 5: We provide all results in summary tables (Tables 1-3), as well as occasionally in figures that supplement these summary tables. Statistical methods as outlined in section 2.7, and we prepared our results section to provide all statistical details recommended by the CONSORT checklist.

Point 6: In my opinion, I think the ‘Conclusions’ and ‘Limitations’ paragraphs would benefit from some thoughtful as well as in-depth considerations by the authors, because as it stands, they are very descriptive but not enough theoretical as discussions should be. Authors should make an effort, trying to explain the theoretical implication as well as the translational application of their research.

Response 6: This manuscript is a companion to the RCT manuscript, which discusses some theoretical point. Our viewpoint is that hypothesizing is useful primarily for definitive, larger studies for which the results are unexpected; this report is neither.

Point 7: References: Authors should consider revising the bibliography, as there are several incorrect citations. Indeed, according to the Journal’s guidelines, they should provide the abbreviated journal name in italics, the year of publication in bold, the volume number in italics for all the references.

Response 7: Our references were prepared using the official MDPI reference style in EndNote, and for all journal articles we provide the abbreviated journal name in italics, the year of publication in bold, the volume number in italics. Some references do have a slightly different format as they are for books, websites, etc, but those references were also formatted using the official MDPI reference style in EndNote.

Reviewer 2 Report

The study is interesting, but I have some considerations to make. In table 1 the full name of the acronyms ADHD and OCS should be inserted.
The sample is very small, especially in relation to the wide age range considered. In the description of the participants it should be reported among the exclusion criteria, if the patients had comorbidities, for example with other neurodevelopment disorders, what was their level of cognition, anxiety or mood disorders, since all these conditions could be associated with tic disorders. It should also be reported if they are on any other therapies during the study, for how long and how often. In fact, all these elements would be useful to better investigate the characteristics of the sample and could influence the results.
The literature should definitely be expanded. It would be interesting to verify, by administering specific standardized tests, whether there is a correlation between the lowering of the level of discomfort of the symptoms and an improvement in the person's adaptive functioning, factors which in neurodevelopmental disorders, which include tic disorders , could contribute to a better quality of life and help counteract the onset of emotional problems that are common. Indeed, recent studies have shown that higher severity levels of symptoms in neurodevelopmental disorders, such as ASD, had less adaptive functioning and in turn the presence of greater adaptive difficulties was related to a greater presence of internalizing problems.

Author Response

Response to Reviewer 2 Comments

Point 1: The study is interesting, but I have some considerations to make. In table 1 the full name of the acronyms ADHD and OCS should be inserted.

Response 1: Thank you for making us aware of this oversight. The acronyms ADHD and OCD are now defined in the table caption.

Point 2: The sample is very small, especially in relation to the wide age range considered.

Response 2: We acknowledge that a larger sample size would be beneficial. However, our pre-registered enrollment plan was to enroll all consenting participants who completed the crossover RCT, for which the planned enrollment was 32. Of these, 31 agreed to participate in this open-label study. This was intended as a small pilot study, as it is the first of its kind. We feel that we gained the useful information we planned to gain.

Point 3: In the description of the participants it should be reported among the exclusion criteria, if the patients had comorbidities, for example with other neurodevelopment disorders, what was their level of cognition, anxiety or mood disorders, since all these conditions could be associated with tic disorders. It should also be reported if they are on any other therapies during the study, for how long and how often. In fact, all these elements would be useful to better investigate the characteristics of the sample and could influence the results.

Response 3: We have included this information in Table 1. For patient comorbidities, we have included both average Y-BOCS scores (a measure of OCD traits) as well as average ADHD rating scale scores. Unfortunately we did not collect data specifically on anxiety or mood disorders. We also include information on current treatments and treatment history. We provide information on the proportion of our sample which is currently taking either antipsychotics or alpha-2 agonists, which are the most common medication treatments for tics. We also provide data on lifetime usage of these medications, as well as lifetime exposure to behavior therapy and other treatments. We feel that this information provides readers with adequate information to understand our results in the context of our particular sample. The full data with all current and past treatments reported by each participant is provided as an online supplement at the links included in the manuscript.

Point 4: The literature should definitely be expanded. It would be interesting to verify, by administering specific standardized tests, whether there is a correlation between the lowering of the level of discomfort of the symptoms and an improvement in the person's adaptive functioning, factors which in neurodevelopmental disorders, which include tic disorders , could contribute to a better quality of life and help counteract the onset of emotional problems that are common. Indeed, recent studies have shown that higher severity levels of symptoms in neurodevelopmental disorders, such as ASD, had less adaptive functioning and in turn the presence of greater adaptive difficulties was related to a greater presence of internalizing problems.

Response 4: This is a very interesting concept, and we agree that the field as a whole would benefit from further exploration on the role of various treatments’ impact on adaptive functioning in tic disorders. However, we collected no such data after treatment, so discussion of this concept would be out of place in this report, which instead focused on the device’s impact on tic frequency and intensity, and on participants’ feedback on the device itself.

Round 2

Reviewer 1 Report

I am very pleased to see that the Authors have welcomed my suggestions and have clarified most of the issues I raised in my first round of this review. I believe that this manuscript entitled ‘Median nerve stimulation for treatment of tics. 2. A 4-week, open trial with ecological momentary assessment’ does an excellent job describing the possible application of median nerve stimulation as a promising treatment for Tourette syndrome. Therefore, I believe that this paper does not need any further revision. 

Thank you for your work,

Reviewer

Reviewer 2 Report

Some of the requested changes have been reported so for me the study is fine.